# DIFF-SHAPE: A NOVEL CONSTRAINED DIFFUSION MODEL FOR SHAPE BASED DE NOVO DRUG DESIGN

## ABSTRACT

Shape-based virtual screening is a widely utilized method in ligand-based de novo drug design, aiming to identify molecules in chemical libraries that share similar 3D shapes but simultaneously possess novel 2D chemical structures compared to the reference compound. As an emerging technology, generative model is an alternative way to do de novo drug design by directly generating 3D novel structures. However, existing models face challenges in reliably generating valid drug-like molecules under specific conformation constrains. Here, a novel diffusion model constrained with 3D reference shape, Diff-Shape, was proposed to generate structures whose 3D conformations are similar to a given reference shape, thereby avoiding the computational cost of screening large database of 3D conformations. This model utilizes a zero-weighted graph control module, taking in various forms of point clouds of reference shape to guide diffusion process of 3D molecular generation. The results show that our model is capable of generating molecules with high shape similarity but still low 2D graph similarity to the query structure and it significantly out-performs existing shape-based generative models.

## 1 INTRODUCTION

Ligand-based drug design (LBDD) task focuses on molecular modelling without relying on knowledge of protein structure. Typically, it elucidates the relationship of a compound's structure and physico-chemical attributes to its biological activity and identifies "actives" on the basis of 3D pharmacophore or 3D shape similarity. For example, 3D shape similarity search is a commonly used ligand-based virtual screening tool to identify molecules with similar shape to a reference structure and has shown promising results for scaffold-hopping tasks.(Rush et al., 2005) However, the effectiveness of virtual screening is tied to the chemical space of searched chemical libraries, restraining its capacity to explore novel chemical space.

In recent years, generative model has emerged as new paradigm for de novo drug design and has revolutionized computer-aided drug design (CADD) by enabling efficient exploration of chemical space and goal-directed molecular optimization (MO) in a data driven manner. Various neural network architectures have been applied in generative models to directly generate drug-like molecules at 1D sequence, 2D graph or 3D conformation level with or without protein structure, including recurrent neural networks (RNN)(Segler et al., 2018; Li et al., 2018; Cho, 2014), variational auto-encoders(Ma et al., 2018; Jin et al., 2020), generative adversarial networks(De Cao & Kipf, 2018), 3D convolutional networks (CNNs)(Kuzminykh et al., 2018), flow-based model(Shi et al., 2020) and, more recently, diffusion based models(Qian et al., 2024; Lin et al., 2022) etc. Among these models, shape conditioned generative model represents an alternative way for doing ligand based de novo design and in contrast to conventional database screening methods, it can explore a much larger chemical space beyond the databases of known chemicals.

The early attempts on shape conditioned generative model were to build models not directly generating 3D conformations. Skalic et al.(Skalic et al., 2019) and Imrie et al.(Imrie et al., 2021) trained networks to generate 1D SMILES strings and 2D molecular graphs, respectively, conditioned on CNN encodings of 3D pharmacophores ignoring Euclidean symmetries. Zheng et al. used supervised molecule-to-molecule translation on SMILES strings for scaffold hopping tasks and evaluated the generated scaffolds' 3D shape similarity to the reference.(Zheng et al., 2021) Papadopoulos et al. sampled molecules with high shape similarity to a target by SMILES based reinforcement learning

in REINVENT, requiring re-optimization of the agent model for each target shape.(Papadopoulos et al., 2021) Roney et al. fine-tuned a 3D generative model on the hits of a shape based virtual screen of more than 1010 drug-like molecules to shift the learned distribution towards a particular shape.(Roney et al., 2022) Yet, this expensive screening approach must be repeated for each new reference shape. The first transferable shape-conditioned 3D generative model, SQUID, proposed by Coely et al., utilized auto-regressive fragment based generation with heuristic bonding geometries, allowing to predict torsion of connecting bonds of growing fragments to make sure the growing 3D conformation properly aligned to the reference shape.(Adams & Coley, 2022) However, its performance was only demonstrated on small drug-like molecules up to 27 atoms. Ning et al. proposed another attempt for shape-constraint molecular generations, ShapeMol(Chen et al., 2023), an equivariant diffusion model conditioned with a shape-embedding from encoder, they demonstrated that ShapeMol out-performed SQUID on shape-conditioned virtual screening.

Recently, various diffusion models have been used for 3D molecule generation in both LBDD and SBDD scenarios, utilizing an iterative denoising process. Welling et al. proposed the equivariant diffusion model (EDM) for 3D molecule generation, improving significantly over previous results in one-shot or auto-regressive settings.(Hoogeboom et al., 2022) The MDM(Huang et al., 2023) and GCDM21(Morehead & Cheng, 2024) modified EDM by limiting the message-passing computations to neighboring nodes and changing a more expressive denoising model. Xu et al. proposed GeoLDM(Xu et al., 2024), a diffusion model on latent space coded by an equivariant autoencoder instead of feature space as defined in EDM. Although these methods have shown promising results, they ignored the connectivity in molecular generations, leading to sub-optimal performance on complex molecules. To overcome this issue, Frossard et al. proposed a mixed graph and 3D denoising diffusion model, MIDI(Vignac et al., 2023), by simultaneously generating a molecular graph and its corresponding 3D coordinates. So far, these diffusion models are conditioned in two ways: conditional guidance existed in the sampling process not in the training stage or using direct control during the training stage with specific model architectures. In former case, an additional end-to-end differential scoring function or model is needed to guide the pre-trained generative model to sample molecules with targeted properties, for example, the KGDiff model by Xu et al.(Qian et al., 2024) and SLIVER model by Mey et al.(Runcie & Mey, 2023) In later case, models such as DiffS-BDD(Schneuing et al., 2022), DiffBP(Lin et al., 2022), and TargetDiff(Guan et al., 2023) were used to generate molecules that bind to a specific protein pocket, the DiffLinker(Igashov et al., 2024) was proposed to generate linkers between molecular fragments. Although these methods improved the conditioned generation in various tasks, all of them need to be trained from scratch for each task which may lead to high computation cost.

In the image generation field, ControlNet(Zhang et al., 2023) achieved success by leveraging the well-established encoding layers of large stable diffusion models, which were pre-trained with billions of images, to generate images under the guidance of input conditions. This approach allowed ControlNet to learn a diverse range of conditional models.(Zhang et al., 2023) Inspired by Control-Net, we introduce a novel diffusion model called Diff-Shape, which combines a pre-trained unconditional diffusion model with a graph control module for shape constrained 3D molecule generation. One unique feature of our method is that the pre-trained unconditional molecule generative model can be directly incorporated into Diff-Shape framework, therefore avoiding the computation cost of training from scratch. By taking in reference shape as point clouds with noise and partially obscured bond information, the graph control module guides diffusion process of molecular generation. To showcase the utility of Diff-Shape in drug design, several tasks of the shape-conditioned generation of chemically diverse molecular structures were highlighted. Our results demonstrate that the Diff-Shape model significantly out-performed existing shape based generative models in terms of the shape similarity of the generation set to the reference shape, and, at the same time, can achieve balanced performance between high 3D shape similarity and relatively low 2D graph similarity which is the key requirement for identification of new scaffold.

## 2 METHODS

In Diff-Shape method, a novel equivariant neural network architecture,named Graph ControllNet (GrCN), was proposed and it composed an unconditioned diffusion model satisfying SE (3) symmetry for 3D molecular generation and a graph control module which takes specific conditions such as the 3D shape of a template molecule. Here we first summarize the architecture of GrCN model,

then how Diff-Shape extends a pre-trained molecular diffusion model, MIDI in this case, to a shape conditioned molecular generative model will be discussed.

## 2.1 GRAPH CONTROLNET

For a given 3D graph $G = \{V, E, R\}$, set V represents node features which include scalar features for all nodes $h_v, \forall v \in V$, such as atom types and charges for atom etc; set $E$ corresponds to edge features including scalar features like bond types for all edges $h_{e_{(v,w)}}, \forall e_{(v,w)} \in E$, and set $R$ refers to atomic coordinates $r_v, \forall v \in V$. The adjacency $Adj$ of $G$ is defined as a matrix where '1' is set for bond existence between two atoms, otherwise is '0'.

Given a trained graph neural network $F(. : \theta)$ with parameters $\theta$, a transformation between two graphs $G_x = \{V_x, E_x, R_x\}$ and $G_y = \{V_y, E_y, R_y\}$ is carried out as:

$$\{V_y, E_y, R_y\} = F(V_x, E_x, R_x, \theta) \tag{1}$$

The scalar features $V_y, E_y$ should be invariant with translation and rotation of $G_x$ in 3D space while the coordinate vector $R_y$ should be equivariant for $F$, thus:

$$\{V_y, E_y, D_y(g)R_y\} = F(\{V_x, E_x, D_x(g)R_x\}, \theta), g \in \mathbb{G} \tag{2}$$

where $\mathbb{G}$ represents a group of transformation operation including transitions and rotations, $D_X(g)$ and $D_Y(g)$ are transformation matrices parameterized by g in $G_x$ and $G_y$.

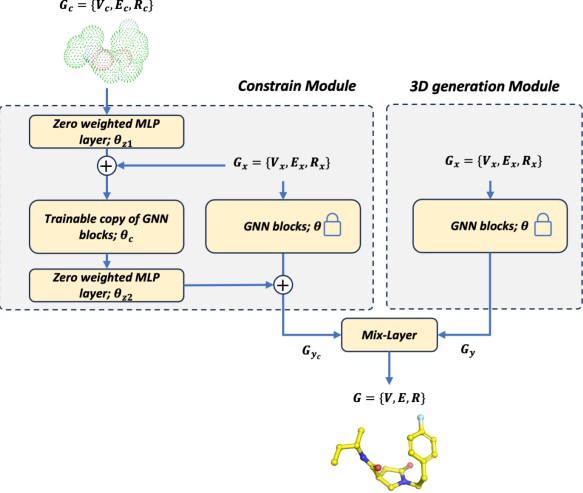

Figure 1: The basic architecture of GrCN, consisting of a constrain module and a 3D generation module.

Inspired by the well-known ControlNet for conditioned image generation, supposing an unconditioned 3D molecule generative model was already trained as a prior model, then GrCN is designed to include a 3D shape constrain module and a 3D molecule generation module as shown in Figure 1, where the 3D generation module is a copy of the prior model with locked parameters $\theta$ and the constrain module contains another locked copy of the prior model and a trainable copy with parameters $\theta_c$. The trainable copy in constrain module takes a condition 3D graph $G_c$ as input. Within the GrCN model, the locked parameters preserve the features in the prior model learned with millions of molecules, while the trainable copy infuses conditioning graph information into prior model to guide the output 3D graph in a profound way.

The trainable copy is connected to the locked blocks with zero-weighted MLP layer, denoted $Z(. \mid .)$, in which all the learnable parameters are initialized to zeros. To build up a GrCN, two instances of

zero-weighted MLP layer with parameters $\theta_{z1}$ and $\theta_{z2}$ are used. The zero-weighted MLP layer actually includes MLPs for scalar features $V, E$ and coordinates $R$. The output of constrain module is shown as:

$$G_{y_c} = F(G_x; \theta) + Z(F(G_x + Z_1(G_c; \theta_{z1}); \theta_c); \theta_{z2}) \tag{3}$$

The final output of GrCN is a linear combination of the constrained module and the unconditioned term as following:

$$G_{out} = G_y + \gamma * (G_{y_c} - G_y) \tag{4}$$

Where $\gamma$ is a scaling factor to control the influence of constrain term on the unconditioned output. In the first training step, since both the weights and bias parameters of zero MLP are initialized to zero, i.e. both of the $Z(. \mid .)$ terms in Eq (3) are zero, then $G_{out} = G_y$. In this way, GrCN enjoys the full capability of the pre-trained prior model, and the harmful noise from conditional graph won't influence the hidden states of trainable copy at the beginning, therefore accelerating and stabilizing the training process.

## 2.2 Diff-Shape Methodology

The goal of Diff-Shape method is to generate 3D molecule conformations with similar 3D shape to a template molecule while try to keep low 2D graph similarity simultaneously. The template molecule is represented as a three-dimensional molecular graph $G_{TM} = \{V_{TM}, E_{TM}, R_{TM}\}$. Several ways of fuzzy operation were applied on $G_{TM}$ to increase its structural obscurity aiming to output solutions having low 2D similarity. As shown in Figure S3, in total, seven different fuzzy operations were adopted in Diff-Shape to achieve varying effects. These operations include: (1) none fuzzy level, in which the original reference molecular graph is used; (2) fuzzy element level, a whitened molecular graph in which all atoms were changed to carbon element and the original bond type information was retained; (3) fuzzy element and bond level, a whitened molecular graph that all bond types are changed to single bond; (4) point cloud level, a whitened three-dimensional point cloud in which all bonds are removed; (5) mixed point cloud level, a whitened molecular graph in which all bonds are changed to single bond and some bonds are partially removed. (6) coloured point cloud level, the original three-dimensional point in which all bonds are removed; and (7) coloured mixed point cloud level, the original element types are kept and other changes are the same with mixed point cloud level. Our result demonstrates that these fuzzy operations play crucial role in generating novel structures compared to the reference structure.

In Diff-Shape, GrCN is implemented as illustrated in Figure 2 for a given time step $t$. A pre-trained MIDI model is employed as the prior model. The algorthim of MIDI for unconditioned generations are briefly introduced in Appendix A1. The embedding blocks of the trainable copy of prior model are utilized to process the condition 3D graph $G_{TM}$, each attaching a zero MLP layer and additionally, two MLPs are used for the scalar features $V, E$ and one E3 MLP for the $R$ set. The E3 MLP can maintain the coordinate equivariance for $R = \{r_0 \ldots r_n\}$ by scaling each $r_i$ relative to its norm $\| r_i \|_2$ as shown in Eq (5).

$$r_i = r_i * MLP(\| r_i \|_2)/(\| r_i \|_2 + \epsilon) \tag{5}$$

Subsequently, the center of geometry is subtracted from the coordinates $R$ of the graph $G$. The graph embedding $G^{t+1} = \{A^{t+1} \oplus C^{t+1}, E^{t+1}, R^{t+1}\}$ is used as the input of time step t and combined with the embedding of $G_{TM}$ to be fed into the trainable module. Two locked copies of prior model are created, one is used in the 3D generation module and the other one is employed in the constraint module.

In the trainable unit of constraint module, a zero initialized MLP layer is attached to each trainable encoder block. Similar to ControlNet, the output of each zero MLP layer is added to the locked GT block of decoder in constraint module. This collecting mechanism is served as skip-connection, linking the output of the trainable encoder layer to the input of the decoder layer in the opposite order. Finally, the graph output of constraint module $G_c = \{V_c, E_c, R_c\}$ is combined with the output of 3D generative module $G_y$ to form the final output of GrCN.

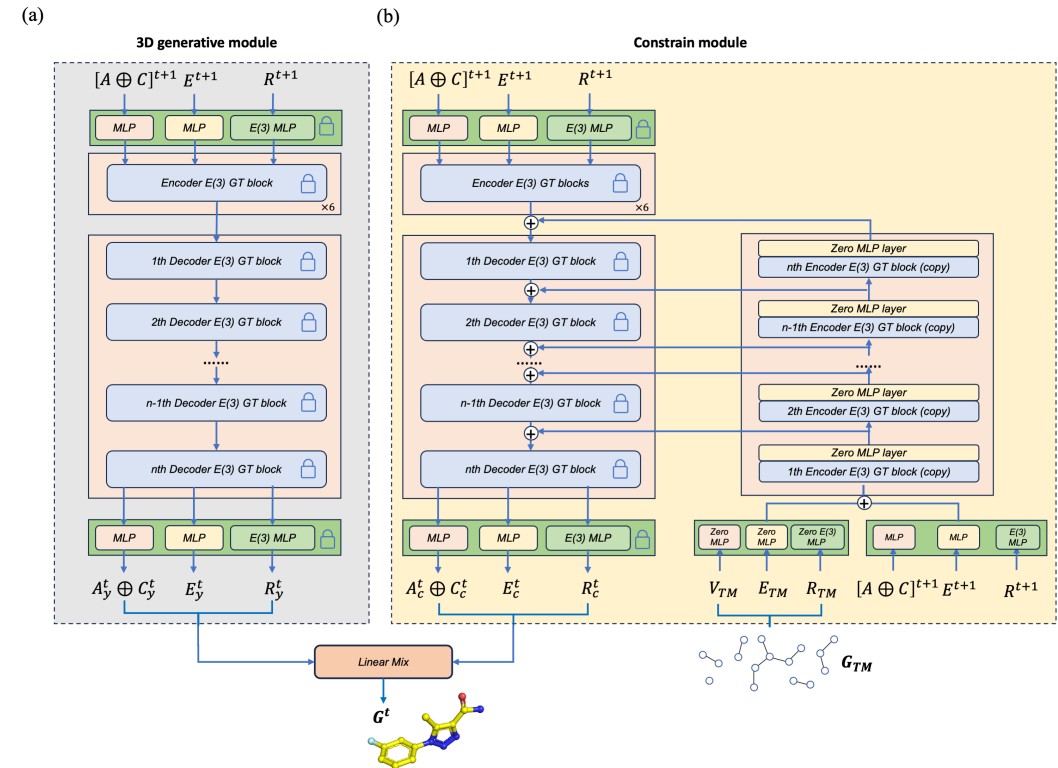

Figure 2: The detailed architecture of Diff-Shape, consisting of (a) 3D generation module and (b) constrain module.

$$G^t = A^t \oplus C^t, E^t, R^t = Mix(A_c \oplus C_c, A_y \oplus C_y), Mix(E_c, E_y), Mix(R_c, R_y) \tag{6}$$

where the Mix function is just a linear combination as shown in Eq (7).

$$Mix(X, Y) = X + \gamma(Y - X) \tag{7}$$

## 2.3 TRAINING AND SAMPLING

For a given molecule $M_0$ represented by $G^0 = \{A^0 \oplus C^0, E^0, R^0\}$ in the training set, during the training phase, the template molecule is itself processed with chosen fuzzy operation as mentioned above and the denoising model is trained to predict the molecule $G^0$ from a noisy input $G^t$ with the shape conditions $G_{TM}$. In the loss function, the estimation of the coordinates $R_\theta$ can simply be optimized with mean-squared error, whereas the prediction $p_\theta^A$ for the atom types, $p_\theta^C$ for the formal charge and $p_\theta^E$ for the bond types corresponds to a classification problem which can be addressed through a cross-entropy loss ($CE$ in the equations). The final training objective of Diff-Shape is a weighted sum of these components:

$$l(G^\theta, p_\theta^G(G^t, G_{TM})) = \lambda_r \parallel R_\theta - R^0 \parallel^2 + \lambda_A CE(A^0, p_\theta^A(G^t, G_R))$$
$$+ \lambda_C CE(C^\theta, p_\theta^C(G^t, G_R)) + \lambda_E CE(E^0, p_\theta^E(G^t, G_R)) \tag{8}$$

When generating new samples, the conditioned posterior $p_\theta$ is defined as the product of denoising models' predictions on each term.

$$p_\theta(G^{t-1} \mid G^t, G_{TM}) =$$
$$\prod_{1 \le i \le n} p_\theta(r_i^{t-1} \mid G^t, G_{TM}) p_\theta(a_i^{t-1} \mid G^t, G_{TM}) p_\theta(c_i^{t-1} \mid G^t, G_{TM}) \prod_{1 \le i,j \le n} p_\theta(e_{ij} \mid G^t, G_{TM})) \tag{9}$$

Similar to MIDI, each term in Eq (9) was calculated by marginalizing over the network prediction, for instance,

$$p_\theta(a_i^{t-1} \mid G^t, G_{TM}) = \sum_{v_i \in V} q(a_i^{t-1} \mid a_i = a, G^t) p_\theta^A(a_i = a \mid G^t, G_{TM}) \tag{10}$$

where $a_i$ is the atom type of node $v_i$, $a$ is the value of $a_i$.

## 2.4 COMPUTATIONAL DETAILS

Diff-Shape was trained on the GEOM-Drugs dataset consisting of 304339 molecules. For each molecule, 5 conformations with lowest energies were selected as our reference datasets. The reference datasets were split into training, valid and test set in a ratio of 8:1:1. For each molecule M in training set, it was trained by using its own shape graph $G_{TM} = \{V_M, E_M, R_M\}$ as the basis of creating shape condition. To increase the transferability of Diff-Shape model, we added noise on both $V_M, E_M$ and $R_M$ with a standard deviation of 0.3 in default, and various fuzzy operations were adopted to construct shape conditions. For fuzzy operations on the mixed point cloud level, the connections between atoms are randomly masked with a ratio of 0.5. The weights of components in loss function on atom type $A$, formal charge $C$, edge $E$ and coordinate $R$ are 0.4, 1.0, 2.0 and 3.0, respectively. The maximal diffusion step was set to 500. Evaluation metrics for Diff-Shape generated molecules includes the 2D graph similarity $Sim_g$ and 3D shape similarity $Sim_{3D}$ with template molecules. The 2D similarity was calculated by the Tanimoto similarity of ECFP-4 fingerprints between generations and templates, while the 3D shape similarity was calculated with the ROCS software of OE Toolkit package in default settings.30(Grant et al., 1996) For the tasks of structure-based drug design, we also evaluate the docking score of generations in target pockets with the GLIDE module of Schrodinger software package.(Halgren et al., 2004) In our study, we utilized an unconditioned MIDI model as the baseline. We conducted a performance comparison between Diff-Shape and two other shape-conditioned generative models: SQUID and ShapeMol. The $\lambda$ parameter of SQUID was set to 0.3 for optimal performance in our practice.(Adams & Coley, 2022) ShapeMol was evaluated with and without guidance using a shape similarity-based score function during the diffusion process to improve its performance. These two variants of Shape-Mol are denoted as "ShapeMol-NoGuide" and "ShapeMol-Guide-0.5", respectively, in which the guidance weight was set to 0.5 in the guidance mode. Furthermore, we evaluated Pocket2Mol and DiffSBDD in the context of structure-based drug design. Both models were employed with their default settings for the evaluation.

## 3 DISCUSSION AND RESULT

### 3.1 COMPOUND QUALITY OF DIFF-SHAPE WITH DIFFERENT 3D SHAPE CONDITIONS

Here, we firstly compared the performance of Diff-Shape with different 3D shape conditions. We randomly selected ten molecules from test set as references to generate 3D molecules. The validity, uniqueness and novelty of 1,000 Diff-Shape generated molecules with 7 types of shape condition are as listed in Table 1 (for mixed point cloud, an additional noise level of 0.4 was used), while the $Sim_g$ and $Sim_{3D}$ are as shown in Figure S4. As a comparison, the unconditioned MIDI generation results are also provided.

The definition of seven shape conditions is listed in Table 1 (also shown in Figure S3). The 'None Fuzzy' condition keep most information of a reference shape including its atom type, bond type and bond connection as well as atomic coordinates, while other conditions miss some information on purpose to make a fuzzy reference shape to achieve balanced performance between high shape

Table 1: The definition of fuzzy operation of shape-condition in Diff-Shape.

| Condition Scheme | Noise level | with atom types | with bond types | with adjacency |
|---|---|---|---|---|
| None fuzzy | | ✓ | ✓ | ✓ |
| Coloured mixed point | | ✓ | ✓ | Mask |
| Coloured point cloud | | ✓ | × | × |
| Fuzzy element | 0.3 | × | ✓ | ✓ |
| Fuzzy element and bond | | × | × | ✓ |
| Mixed point cloud | | × | × | Mask |
| Point cloud | | × | × | × |
| Mixed point cloud | 0.4 | × | × | Mask |

Table 2: The performance of Diff-Shape models trained with different fuzzy operations of shape-condition.

| Scheme index | Noise level | Validity | Uniqueness | Validity* Uniqueness | Connected components | Molecule Stable |
|---|---|---|---|---|---|---|
| None fuzzy | | 0.940 | 0.050 | 0.047 | 0.998 | 0.94 |
| Coloured mixed point cloud | | 0.837 | 0.144 | 0.121 | 0.997 | 0.88 |
| Coloured point cloud | | 0.752 | 0.245 | 0.184 | 0.995 | 0.84 |
| Fuzzy element | 0.3 | 0.882 | 0.159 | 0.140 | 0.978 | 0.99 |
| Fuzzy element and bond | | 0.908 | 0.253 | 0.230 | 0.998 | 0.94 |
| Mixed point cloud | | 0.806 | 0.491 | 0.396 | 0.984 | 0.88 |
| Point cloud | | 0.629 | 0.903 | 0.568 | 0.947 | 0.76 |
| Mixed point cloud (0.4) | 0.4 | 0.634 | 0.920 | 0.583 | 0.953 | 0.77 |
| MIDI | - | 0.769 | 1.000 | 0.769 | 0.902 | 0.91 |

similarity and low 2D graph similarity. The general compound qualities for generations can be found in Table 2. 'None Fuzzy' shape condition results in less unique but more valid structures. In contrast, applying less precise (fuzzier) condition leads to more unique but less valid structures. We examine the multiplication of validity and uniqueness as a balanced metric score. Among seven fuzzy levels of shape condition, point cloud level generates the most valid and unique molecules with a value of around 0.57 and the order is 'Point cloud' > 'Mixed point cloud' > 'Fuzzy element and bond' > 'Coloured point cloud' > 'Fuzzy element' > 'Coloured mixed point cloud'> 'None fuzzy'. However, once we add more noise (noise level of 0.4) on the mixed point cloud model, it (with the core of 0.583) out-performed the point cloud model. The shape similarity and 2D graph similarity of generations under various shape conditions are displayed in Figure S4. In general, the stricter shape condition used, the higher 2D similarity is between the generated compound set and the reference. For the fuzziest option 'Mixed Point cloud' with noise of 0.4, its median shape similarity is 0.823, while the median 2D similarity is 0.187, which means that our model is capable in generating compounds with high shape similarity while keeping low 2D similarity. It is worthwhile to note that for mixed point cloud models under noise level of 0.3 and 0.4, the fuzzier model (noise level of 0.4) tends to generate more novel structures, while its shape similarity decreases slightly. Given these results, mixed point cloud model was chosen for doing following experiments in current study.

### 3.2 THE PERFORMANCE OF DIFF-SHAPE ON SHAPE-CONDITIONED STRUCTURE GENERATION

We then evaluated Diff-Shape's performance in the shape-based virtual screening scenario by comparing 3D shape similarity and 2D graph similarity of generations. For comparison, we examined the performance of unconditioned MIDI model as baseline and other shape-conditioned models,

SQUID, ShapeMol-NoGuide and ShapeMol-Guide-0.5 in our case. The evaluation was performed on randomly selected 100 molecules from the test set as the reference set. For each reference molecule, 100 valid and unique molecules were generated and assessed. The validity, uniqueness as well as similarity scores of generations are shown in Table 3. Despite Diff-Shape generations showing lower validity and uniqueness than SQUID, and ShapeMol models, it significantly outperformed SQUID, ShapeMol and MIDI in terms of identifying molecules with $Sim_{3D} > 0.8$. Specifically, Diff-Shape achieved a ratio of 0.91 and 0.66 for such molecules with noise level of 0.3 and 0.4 respectively, while SQUID, ShapeMol-NoGuide, ShapeMol-Guide-0.5 and MIDI can only achieve 0.127, 0.001, 0.003 and 0.002, respectively. This highlights that Diff-Shape model does a better job in generating compounds with similar shape, while the drawback of lower validity and uniqueness can be largely solved by sampling larger number of compounds and filtering invalid ones.

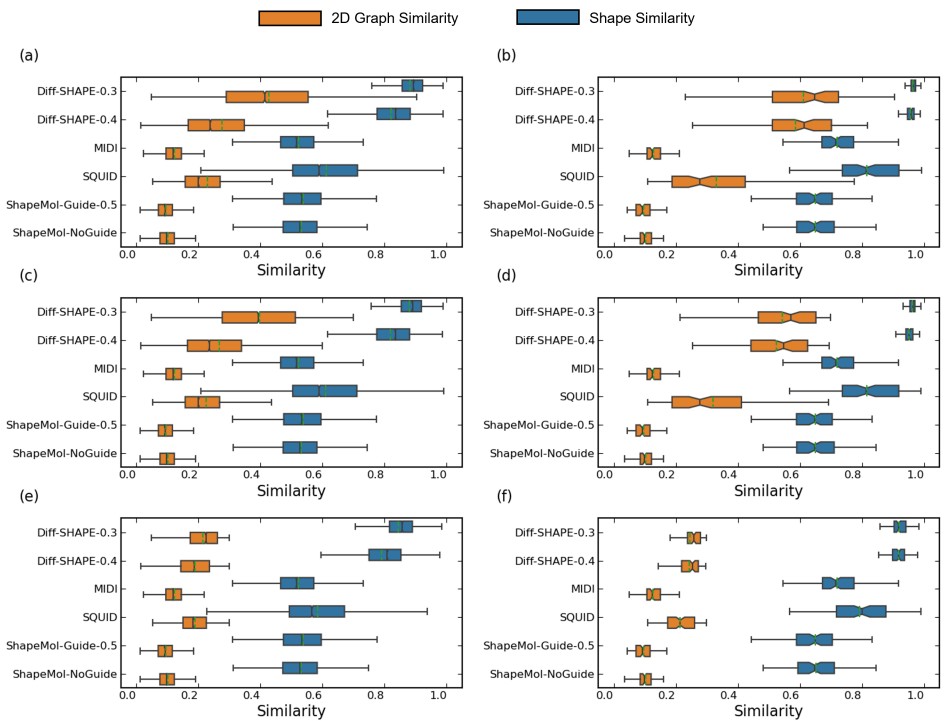

Figure 3: The 3D shape similarity $Sim_{3D}$ and 2D graph similarity $Sim_g$ distribution of the generated molecules for 100 references. The performance of all the generations with graph similarity less than 1.0 (a), 0.7(c) and 0.3 (e); The performance of the best generations with graph similarity less than 1.0 (b), 0.7 (d) and 0.3 (f). The green dash lines are the average values of distribution.

Considering 2D similarity of the generated set to the reference compound, we filtered molecules by thresholds of 2D similarity $Sim_g$ with 1.0, 0.7, and 0.3 to gauge the level of novelty in generations regarding to the reference. The results in Table 3 show that the ratio of compound whose 3D shape similarity is larger than 0.8 and 2D similarity is less than 0.7 are 0.830 and 0.641 for Diff-Shape with noise level of 0.3 and 0.4, 0.123 for SQUID, 0.003 and 0.001 for ShapeMol with/without guidance. For compounds whose 3D shape similarity is larger than 0.8 and 2D similarity is less than 0.3, the ratio is 0.213 and 0.353 for Diff-Shape with noise of 0.3 and 0.4, 0.055 for SQUID, 0.003 and 0.001 for ShapeMol with/without guidance. This again demonstrates our method has better chance in generating more diverse compounds with similar shape, comparing with other shape conditioned generation models.

The shape similarity distributions were analyzed for both the entire generated sets and the best sets, i.e. molecules with the highest shape similarity to each reference. At the same time, we

Table 3: The performance of generations on 100 reference molecules.

| Method | Validity | Uniqueness | Validity* Uniqueness | $P_{Sim_{3D}>0.8}^{Sim_g<1.0}$ | $*P_{Sim_{3D}>0.8}^{Sim_g<0.7}$ | $P_{Sim_{3D}>0.8}^{Sim_g<0.3}$ |
|---|---|---|---|---|---|---|
| Diff-Shape-0.3[a] | 0.831 | 0.515 | 0.413 | 0.913 | 0.830 | 0.213 |
| Diff-Shape-0.4[b] | 0.629 | 0.946 | 0.591 | 0.657 | 0.641 | 0.353 |
| MIDI | 0.769 | 1 | 0.769 | 0.002 | 0.002 | 0.002 |
| SQUID | 0.995 | 0.937 | 0.932 | 0.127 | 0.123 | 0.055 |
| ShapeMol-NoGuide | 0.983 | 0.999 | 0.983 | 0.001 | 0.001 | 0.001 |
| ShapeMol-Guide-0.5 | 0.980 | 1.000 | 0.980 | 0.003 | 0.003 | 0.003 |

[a] with noise level of 0.3;
[b] with noise level of 0.4;
* ration of molecules with $Sim_{3D} > 0.8$ to reference and $Sim_g < 1.0$.

also examined the shape similarity distribution according to 2D similarity cut-off 1.0, 0.7, 0.3. For compounds with $Sim_g < 1.0$, the median of $Sim_{3D}$ was 0.91 among all generations and 0.96 for the best performers for Diff-Shape-0.3.

Figure 4: The shape alignment of two-best generations with $Sim_g < 0.3$ to six reference molecules. The dotted surfaces represent the 3D shape of references.

Analysis results for compounds with $Sim_g < 0.7$ are described in Figure 3c and 3d, while 3e and 6f for compounds with $Sim_g < 0.3$. Remarkably, even for low 2D similarity molecules ($Sim_g < 0.3$),

their 3D shape similarities $Sim_{3D}$ are only decreased slightly for Diff-Shape-0.3, which are 0.84 and 0.91 for all generations and the best ones, respectively (Figure 3e and 3f). This is significantly better than other compared models. For Diff-Shape-0.4, the slight increase in noise level leads to a noteworthy improvement in the diversity of generation, with only a minor decrease in 3D shape similarity for all generations as well as the best generations. This suggests that users can control the diversity of generations by adjusting the noise level.

Although $Sim_g$ of both ShapeMol generations are less than 0.2 in general, their median $Sim_{3D}$ were around 0.55 for all generations, which is only marginally better than the baseline MIDI model. It is even worse for the best ones with a median of 0.64, compared to the 0.71 for MIDI. The SQUID model performed better than ShapeMol models, the median $Sim_{3D}$ of SQUID generations are 0.58 and 0.83 for all generations and the best ones. Nevertheless, this is still significantly lower than those of Diff-Shape. The shape alignments of the two best generations with $Sim_g < 0.3$ to the reference molecules are displayed in Figure 4, where Diff-Shape's generations achieved alignment with $Sim_{3D} > 0.80$ for all six references, showing considerable good overlap with the shape conditions, whereas SQUID failed on the 2nd, 4th, and 6th references, ShapeMol models only succeed in 1st reference with the guidance, and MIDI failed in all cases. These findings demonstrate Diff-Shape's potential in shape-based ligand virtual screening.

## 4 CONCLUSION

In current study, we introduce a novel diffusion-based 3D molecular generative model, called as Diff-Shape, for shape-controlled 3D molecule generation. By directly incorporating a pre-trained unconditional 3D molecular generative model and a graph control module taking reference shape as input, the condition guided diffusion process of molecular generation can be achieved. In tasks of the shape-conditioned generations of chemically diverse molecular structures, our results demonstrate that the Diff-Shape model can efficiently generate drug-like molecular conformations which has not only high 3D similarity to the reference shape but also possess reasonable chemical novelty compared to the reference structure.

### AVAILABILITY

The source code and experimental data of Diff-Shape are available from `https://github.com/4science0/Diff-shape`.

### COMPETING INTERESTS

The Authors declare no Competing Financial or Non-Financial Interests.

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

# A APPENDIX

## A.1 UNCONDITIONAL 3D MOLECULE GENERATIVE MODEL

In GrCN, a diffusion-based model, MIDI, was used for unconditional 3D molecule generation. Here, we briefly introduce its algorithm, details should be referred to the original reference.(Vignac et al., 2023) Given an input molecule $M_0$, represented by $G_0 = \{A_0 \oplus C_0, E_0, R_0\}$, where $A \oplus C$ is concatenation of one-hot encoding of atom type and formal charge for atoms, $E_0$ contains the bond types, while the set R refers to the atom coordinates. MIDI corrupts the features of each node and edge independently with an adaptive noise schedule. The node and edge features $V_0, E_0$ are diffused discretely, where noise model is a sequence of categorical distribution C and the coordinates R is diffused with gaussian noise $\epsilon$ within the zero center-of-mass (CoM) range, where $\epsilon \sim N^{CoM}(\alpha^t R^{t-1}, (\sigma^t)^2 I)$, the parameter $\alpha^t_{t \leq T}$ controls how much signal is retained at each step and $\sigma^t_{t \leq T}$ indicates how much noise is added. Then, the final noise model is given by:

$$q(G^t \mid G^{t-1}) \sim N^{CoM}(\alpha^t R^{t-1}, (\sigma^t)^2 I) \times C(A^{t-1} Q_A^t) \times \mathcal{C}(C^{(t}-1)Q_C^t) \times \mathcal{C}(E^{(t}-1)Q_E^t) \quad \text{(S1)}$$

where $Q_A^t$, $Q_C^t$ and $Q_E^t$ are transition matrices for the categorical distributions to $A,C$ and $E$, as described in ref of MIDI.(Vignac et al., 2023)

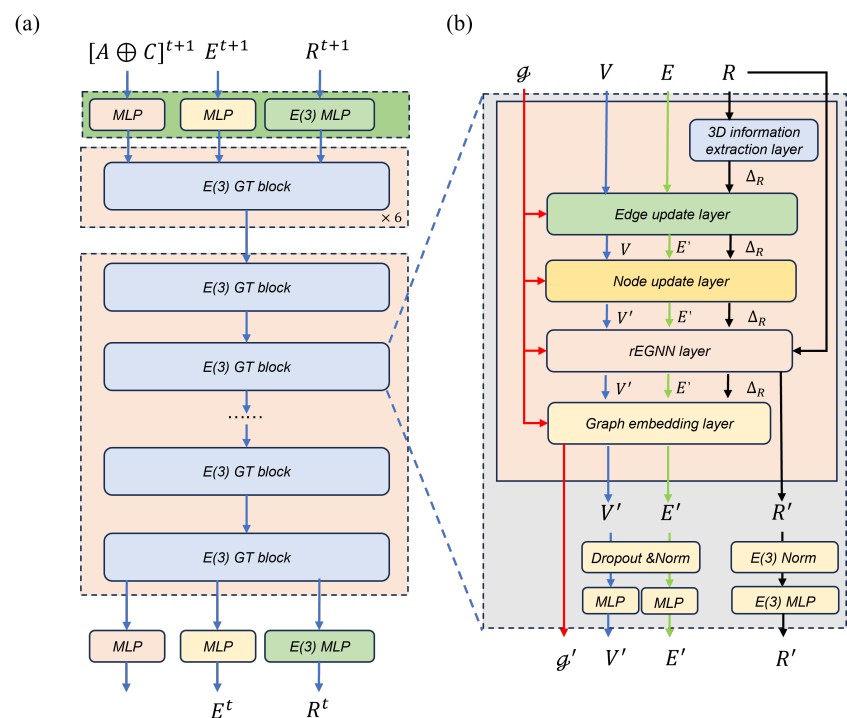

Figure S1: The architecture of MIDI denoising model (a) and its basic unit of E(3) graph transformer blocks (b).

The denoising model of MIDI is a Transformer model, consisting of 12 equivariant graph transformer (E3-GT) blocks, as depicted in Figure S1a. The first six of these blocks function as the encoder, and the latter half as the decoder. As demonstrated in Figure S1a, each E3-GT block processes 3D molecular graph data $G = \{A, C, E, R, \}\}$, and performs a sequence of internal operations to update the graph's features. These operations include the extraction of 3D information, updating of edge ($E$) and node features ($V = A \oplus C$), R coordinates with rEGNN(Satorras et al., 2021),and graph embeddings ({}), followed by drop-out and normalization steps, which are detailed in Figure S2 and ref of MIDI.(Vignac et al., 2023) The coordinate updating relies on distances between atoms, which preserves the E3 equivariance of the graph.

When generating new samples, the posterior $p_\theta$ is defined as the product of denoising models' predictions on each term.

$$p_\theta(G^t \mid G^{t+1}) = \prod_{1 \leq i \leq n} p_\theta(r_i^t \mid G^{t+1}) p_\theta(a_i^t \mid G^{t+1}) p_\theta(c_i^t \mid G^{t+1}) \prod_{1 \leq i,j \leq n} p_\theta(e_{ij}^t \mid G^{t+1}) \quad \text{(S2)}$$

Each term in Eq (S2) is calculated by marginalizing over the network prediction, for instance:

$$p_\theta(a_i^t \mid G^{t+1}) = \sum_{v_{i \in V}} q(a_i^t \mid a_i = a, G^{t+1}) p_\theta^A(a_i = a) \quad \text{(S3)}$$

Where $a_i$ is the atom type of node $v_i$, $a$ is the value of $a_i$.

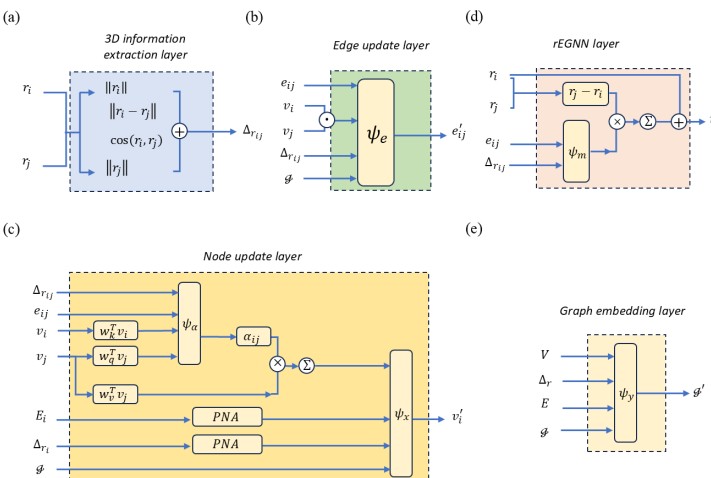

Figure S2: The detailed architectures of MIDI's 3D information extraction layer (a), edge update layer (b), node update layer (c), rEGNN layer (d), graph embedding layer (e).

## A.2 FUZZY OPERATIONS ON SHAPE CONDITION

As shown in Figure S3, in total, seven different fuzzy operations were adopted in Diff-Shape to achieve varying effects. These operations include: (1) none fuzzy level, in which the original reference molecular graph is used; (2) fuzzy element level, a whitened molecular graph in which all atoms were changed to carbon element and the original bond type information was retained; (3) fuzzy element and bond level, a whitened molecular graph that all bond types are changed to single bond; (4) point cloud level, a whitened three-dimensional point cloud in which all bonds are removed; (5) mixed point cloud level, a whitened molecular graph in which all bonds are changed to single bond and some bonds are partially removed. (6) coloured point cloud level, the original three-dimensional point in which all bonds are removed; and (7) coloured mixed point cloud level, the original element types are kept and other changes are the same with mixed point cloud level. Our result demonstrates that these fuzzy operations play crucial role in generating novel structures compared to the reference structure.

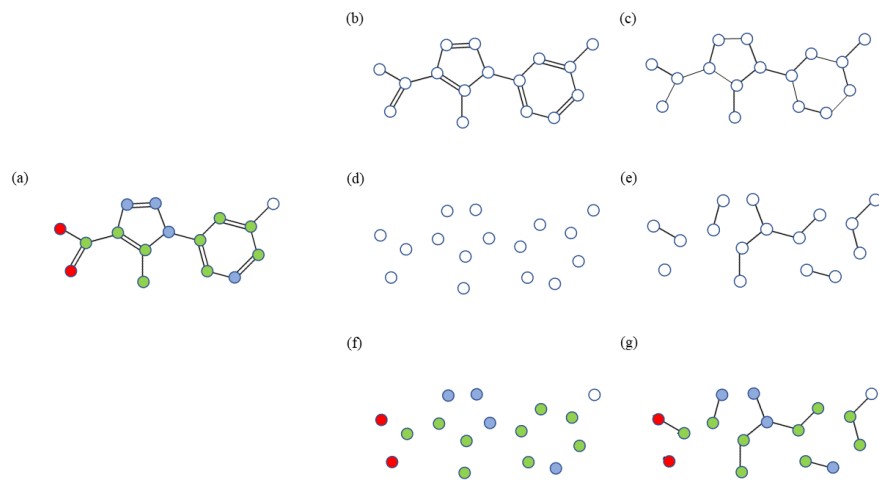

Figure S3: Seven types of shape conditions as input of constrain module including (a) none fuzzy, (b) fuzzy element, (c) fuzzy element and bond, (d) point cloud, (e) mixed point cloud, (f) coloured point cloud and (g) coloured mixed point cloud.

## A.3 PERFORMANCE OF DIFF-SHAPE WITH DIFFERENT 3D SHAPE CONDITIONS (FUZZY OPERATIONS)

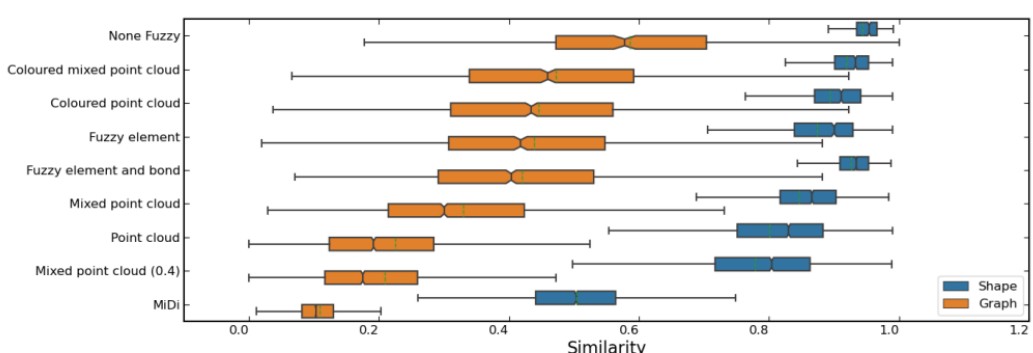

Figure S4: The distributions of 3D shape similarity $Sim_{3D}$ and 2D graph Tanimoto similarity $Sim_g$ of Diff-Shape models trained with different fuzzy operations of shape condition. The green dashed lines represent average values.

