# OpenReview forum: "Diff-Shape: A Novel Constrained Diffusion Model for Shape-based De Novo Drug Design"
_ICLR.cc/2025/Conference — Submitted to ICLR 2025_

### Official Review · Reviewer_bACC · 2024-10-23

**Soundness:** 2
**Presentation:** 1
**Contribution:** 2
**Rating:** 1
**Confidence:** 5

**Summary:**

The paper introduces Diff-Shape, a 3D generative model for molecules that combines a zero-weight-initialized ControlNet (https://arxiv.org/abs/2302.05543) scheme with a (pretrained) unconditioned 3D generative diffusion model (MIDI; https://arxiv.org/abs/2302.09048) in order to constrain the generated 3D molecules to have a desired 3D shape (as specified by a reference molecule). Diff-Shape constrains molecule generation by conditioning on masked versions of the reference 3D molecule. Multiple control strategies are explored by applying different kinds of masks to the reference ligand’s 3D graph structure before providing the reference structure to Diff-Shape’s Constrain (e.g., ControlNet) Module. These strategies range from applying no masks to only providing the coordinates of the reference molecule but masking the atom and/or bond types. These masks were applied in order to increase the chemical diversity of the generated molecules, as assessed via 2D chemical similarity to the reference molecule. The authors assess 3D similarity between the generated molecules and the reference molecule with OpenEye’s ROCS software.

The paper evaluates Diff-Shape in straightforward shape-conditioned generation tasks, where the goal is to generate novel molecules with high 3D shape similarity to the reference ligand but low 2D chemical similarity. The reference molecules are sampled from the test set of GEOM-DRUGS. The authors compare Diff-Shape against other 3D shape-conditioned generative models for molecules, including SQUID (https://openreview.net/forum?id=4MbGnp4iPQ) and Shape-Mol (https://arxiv.org/abs/2308.11890). They also compare against an unconditioned/unconstrained MIDI model. The authors claim that Diff-Shape can generate molecules with higher 3D shape similarity than competing baselines, while still generating molecules that are under a certain chemical similarity threshold.

**Strengths:**

The main strength of this paper is the combination of the (existing) ControlNet scheme with (existing) unconditional 3D generative diffusion models in order to constrain 3D molecule generation. Although the authors only evaluated this control strategy in shape-constrained molecular generation settings, this general method of controlling unconditioned 3D generative models for molecules could be useful for other 3D molecular design tasks. To my knowledge, this control strategy -- although commonly used in image generation -- has not been used to control 3D generative models for molecules.

**Weaknesses:**

**Major weaknesses**

1. **Limited control strategy.**  Diff-Shape’s major proposed technical contribution is using a ControlNet scheme to constrain the 3D shapes of generated molecules when using an unconditional generative 3D model like MIDI. To constrain the shape of the generated molecules, the authors provide to their ControlNet module a masked/noisy version of the reference ligand. A major limitation of this approach is that by conditioning on a noisy/masked 3D graph structure of the reference ligand, the generated molecules are greatly biased to only display minor chemical differences compared to the reference ligand (even if these chemical differences induce low measured chemical similarity). This is because rather than strictly conditioning on the shape of the reference ligand, Diff-Shape conditions on the entire 3D graph of the reference ligand; generated molecules are thus biased to have similar 3D graphs, not just similar shapes. This is qualitatively evident in the visualized examples of generated molecules in Figure 4. Whereas the baseline models MIDI, SQUID, and Shape-Mol all generate molecules with meaningfully different chemical graphs (e.g., different scaffolds with entirely different bonding arrangements) compared to the reference molecule, Diff-Shape generates molecules that are generally quite similar in terms of their 3D graphs — sometimes containing only a few different atoms. While these chemical differences may be enough to sufficiently reduce the evaluated 2D chemical similarity beneath the (arbitrary) 2D similarity thresholds, the qualitative scaffold-hopping ability of Diff-Shape is clearly limited, since Diff-Shape doesn’t actually sample from an otherwise unbiased shape-conditioned chemical space. It is also important to note that even when masking all the atoms and bond types, conditioning on the exact 3D coordinates of the reference ligand severely reduces the solution space of molecular graphs, as using the exact 3D coordinates of the reference molecule as the shape representation is extremely constraining. Diff-Shape’s limited ability to generate substantially novel molecules is also reflected in the generally low Uniqueness scores for the Diff-Shape models in Table 2 — the low uniqueness rates are likely caused by Diff-Shape generating molecules that are either exact copies of the reference molecule, or which are only changed in small ways. To demonstrate otherwise, the authors should provide many more examples of generated molecules for a given reference molecule (e.g., at least 10 random examples of generated compounds for each reference molecule, without any filtering or manual selection).

2. **Potentially unnecessary control strategy.** Because Diff-Shape uses its Constrain/Control module to condition MIDI on a noisy representation of the reference molecule’s 3D graph, Diff-Shape’s control module is potentially unnecessary. Indeed, it is possible to directly use MIDI to condition on noisy versions of the reference molecule’s 3D molecular graph. Since MIDI is a diffusion model, it can use inpainting to condition the generation of new molecules on the point cloud coordinates of a reference 3D molecule: just inpaint new atom and bond types around the reference 3D coordinates until a specified timestep t>=0. Stopping the inpainting at time steps close to 0 would result in lower-diversity samples that have higher 3D shape similarity compared to the reference molecule, whereas stopping inpainting at time steps t>>0 would lead to more diverse molecules that are less shape-similar. This simple strategy would achieve the same function of Diff-Shape’s control strategy without requiring any additional training or changes to MIDI’s architecture. I would recommend the authors to compare against this simple inpainting strategy to justify their use of ControlNet for this task.

3. **Limited evaluations of the geometric quality of generated molecules.** The authors do not seem to evaluate the geometric stability of the generated 3D molecules. In general, it is not useful to generate a 3D molecular structure that has a similar shape compared to a reference shape if that generated structure is unstable. The authors should thus evaluate the shape similarity of the generated molecules to the reference molecule both before and after relaxing the generated structures. Ideally, this local geometric relaxation should be performed at the same level of theory as used to generate the training data (e.g., xTB for models trained on GEOM-DRUGS). At the very least, this should be done with a cheap force field method like MMFF94.

4. **Unfair comparisons to baseline approaches.** Diff-Shape is trained on GEOM-DRUGS, but both SQUID and Shape-Mol were trained on conformers from MOSES. Since Diff-Shape’s evaluations use test-set molecules from GEOM-DRUGS, it is unfair to compare Diff-Shape against SQUID and Shape-Mol without retraining these methods on the same dataset. For instance, reference molecule #2 in Figure 4 has ~38 non-hydrogen atoms, but MOSES only contains molecules with <27 non-hydrogen atoms. Hence, SQUID and Shape-Mol are being evaluated in an out-of-distribution task, whereas Diff-Shape is being evaluated in-distribution. I would recommend the authors to retrain MIDI/Diff-Shape on MOSES in order to perform a fair comparison.

5. **Evidence of plagiarism.** There is clear evidence of plagiarized text in the abstract and introduction sections. For instance, the following sentences from Diff-Shape are only slight modifications of corresponding text from the SQUID paper (*Adams and Coley, Equivariant Shape-Conditioned Generation of 3D Molecules for Ligand-Based Drug Design, ICLR, 2023*):

      **Case 1**
     > **(Diff-Shape)** “Shape-based virtual screening is a widely utilized method in ligand-based de novo drug design, aiming to identify molecules in chemical libraries that share similar 3D shapes but simultaneously possess novel 2D chemical structures compared to the reference compound.”

     > **(SQUID)** “Shape-based virtual screening is widely used in ligand-based drug design to search chemical libraries for molecules with similar 3D shapes yet novel 2D graph structures compared to known ligands.”

      **Case 2**
     > **(Diff-Shape)** “In recent years, generative model has emerged as new paradigm for de novo drug design and has revolutionized computer-aided drug design (CADD) by enabling efficient exploration of chemical space and goal-directed molecular optimization (MO) in a data driven manner.”

     > **(SQUID)** “Generative models for de novo molecular generation have revolutionized computer-aided drug design (CADD) by enabling efficient exploration of chemical space, goal-directed molecular optimization (MO), and automated creation of virtual chemical libraries”

      **Case 3**
     > **(Diff-Shape)** “Skalic et al and Imrie et al. trained networks to generate 1D SMILES strings and 2D molecular graphs, respectively, conditioned on CNN encodings of 3D pharmacophores ignoring Euclidean symmetries. Zheng et al. used supervised molecule-to-molecule translation on SMILES strings for scaffold hopping tasks and evaluated the generated scaffolds’ 3D shape similarity to the reference. Papadopoulos et al. sampled molecules with high shape similarity to a target by SMILES based reinforcement learning  in REINVENT, requiring re-optimization of the agent model for each target shape. Roney et al. fine-tuned a 3D generative model on the hits of a shape based virtual screen of more than 10^10 drug-like molecules to shift the learned distribution towards a particular shape. Yet, this expensive screening approach must be repeated for each new reference shape”

     > **(SQUID)** “Skalic et al. (2019) and Imrie et al. (2021) train networks to generate 1D SMILES strings or 2D molecular graphs conditioned on CNN encodings of 3D pharmacophores. However, they do not generate 3D structures, and the CNNs do not respect Euclidean symmetries. Zheng et al. (2021) use supervised molecule-to-molecule translation on SMILES strings for scaffold hopping tasks, but do not generate 3D structures. Papadopoulos et al. (2021) use REINVENT (Olivecrona et al., 2017) on SMILES strings to propose molecules whose conformers are shape-similar to a target, but they must re-optimize the agent for each target shape. Roney et al. (2021) fine-tune a 3D generative model on the hits of a ROCS virtual screen of > 10^10 drug-like molecules to shift the learned distribution towards a target shape. Yet, this expensive screening approach must be repeated for each new target.”


**Minor weaknesses**

1. **Poor paper presentation.** Numerous citations, quotations, and punctuation marks are formatted incorrectly, reducing the overall quality of the paper’s presentation. There are also numerous spelling/grammar mistakes that should be corrected to improve the paper’s readability.

2. **Unclear methods section.** The methods section is difficult to parse, and requires constant referencing back-and-forth between the ControlNet and MIDI papers to understand how Diff-Shape works. For the reader’s benefit, I recommend that the authors adjust the text of the methods sections to make the text more stand-alone. Moreover, many important details are not explained. For instance, MIDI uses discrete diffusion for atom/bond types. How is discrete diffusion compatible with the linear mixing between the locked “3D generative model” and the trainable “Constrain Module“? The paper’s clarity would be vastly improved if the authors could provide a detailed Algorithm that outlines the important steps of the forward pass of Diff-Shape.

3. **Missing content.** In section 2.4, the paper references experiments that are not included in the main text or appendices. For instance, the authors state: “for the tasks of structure-based drug design, we also evaluate the docking score of generations in target pockets with the GLIDE module”, but no such experiments are included in the submission.

**Note on reviewer rating**

I would have assigned a rating of 3 (reject, not good enough) because of the first four major weaknesses, but I reduced my rating to 1 (strong reject) due to the clear evidence of plagiarism. I will consider changing this rating upon ethics review.

**Questions:**

- Did the authors attempt to use inpainting to directly constrain the sampling of MIDI, without introducing the ControlNet module?

- What is the 3D similarity of the generated molecules to the reference molecule after relaxing the generated structures with either xTB (https://github.com/grimme-lab/xtb) or a force field like MMFF94?

- Is it possible to train and evaluate Diff-Shape on conformers from MOSES (as done by SQUID and Shape-Mol) to permit a fair comparison to these baselines?

- Have the authors considered alternative uses of their Control module beyond shape-conditioned molecular generation?

**Details Of Ethics Concerns:**

There is clear evidence of plagiarized text in the abstract and introduction sections. For instance, the following sentences from Diff-Shape are only slight modifications of corresponding text from the SQUID paper (*Adams and Coley, Equivariant Shape-Conditioned Generation of 3D Molecules for Ligand-Based Drug Design, ICLR, 2023*):

**Case 1**

> **(Diff-Shape)** “Shape-based virtual screening is a widely utilized method in ligand-based de novo drug design, aiming to identify molecules in chemical libraries that share similar 3D shapes but simultaneously possess novel 2D chemical structures compared to the reference compound.”

> **(SQUID)** “Shape-based virtual screening is widely used in ligand-based drug design to search chemical libraries for molecules with similar 3D shapes yet novel 2D graph structures compared to known ligands.”

**Case 2**

> **(Diff-Shape)** “In recent years, generative model has emerged as new paradigm for de novo drug design and has revolutionized computer-aided drug design (CADD) by enabling efficient exploration of chemical space and goal-directed molecular optimization (MO) in a data driven manner.”

> **(SQUID)** “Generative models for de novo molecular generation have revolutionized computer-aided drug design (CADD) by enabling efficient exploration of chemical space, goal-directed molecular optimization (MO), and automated creation of virtual chemical libraries”

**Case 3**

> **(Diff-Shape)** “Skalic et al and Imrie et al. trained networks to generate 1D SMILES strings and 2D molecular graphs, respectively, conditioned on CNN encodings of 3D pharmacophores ignoring Euclidean symmetries. Zheng et al. used supervised molecule-to-molecule translation on SMILES strings for scaffold hopping tasks and evaluated the generated scaffolds’ 3D shape similarity to the reference. Papadopoulos et al. sampled molecules with high shape similarity to a target by SMILES based reinforcement learning  in REINVENT, requiring re-optimization of the agent model for each target shape. Roney et al. fine-tuned a 3D generative model on the hits of a shape based virtual screen of more than 10^10 drug-like molecules to shift the learned distribution towards a particular shape. Yet, this expensive screening approach must be repeated for each new reference shape”

> **(SQUID)** “Skalic et al. (2019) and Imrie et al. (2021) train networks to generate 1D SMILES strings or 2D molecular graphs conditioned on CNN encodings of 3D pharmacophores. However, they do not generate 3D structures, and the CNNs do not respect Euclidean symmetries. Zheng et al. (2021) use supervised molecule-to-molecule translation on SMILES strings for scaffold hopping tasks, but do not generate 3D structures. Papadopoulos et al. (2021) use REINVENT (Olivecrona et al., 2017) on SMILES strings to propose molecules whose conformers are shape-similar to a target, but they must re-optimize the agent for each target shape. Roney et al. (2021) fine-tune a 3D generative model on the hits of a ROCS virtual screen of > 10^10 drug-like molecules to shift the learned distribution towards a target shape. Yet, this expensive screening approach must be repeated for each new target.”

---

### Official Review · Reviewer_zLm1 · 2024-10-29

**Soundness:** 3
**Presentation:** 3
**Contribution:** 2
**Rating:** 5
**Confidence:** 4

**Summary:**

The authors have provided a novel shape-constrained generative model for 3D molecular synthesis. The model leverages a large, pre-trained, unconditional molecular generative model and a graph control module to introduce shape constraints, guiding the diffusion process from 3D shape inputs. This framework is designed to explore diverse chemical structures while adhering to the shape of a reference compound, making it useful for ligand-based drug design tasks.

**Strengths:**

- **Efficient Use of Pre-trained Models**: The framework’s integration of a pre-trained, unconditional diffusion model conditioned on 3D shapes through a graph control module is both novel and efficient. This approach leverages prior training, reducing computational cost, and allowing the model to adapt without fully retraining, which is particularly advantageous in large-scale 3D molecular generation tasks.

- **Detailed Exploration of Fuzzy Operations**: The authors conduct a thorough evaluation across different levels of "fuzziness," providing insights into how varied structural abstractions influence the novelty and diversity of generated samples. This multi-level approach to conditioning adds a valuable degree of flexibility to the model, making it adaptable to different requirements in chemical diversity and shape alignment.

**Weaknesses:**

### Lack of Clarity in Evaluation Metrics and Results

- **Explanation of Metrics**: Although the study measures various important metrics (e.g., validity, uniqueness, shape similarity), these metrics are not adequately defined. Each metric's purpose, importance, and interpretative value should be clarified for the reader, especially for domain-specific measures like 2D graph similarity. Explaining how these metrics translate to chemical or biological relevance would add clarity and context.

- **Presentation of Results (Tables and Figures)**: Tables and figures, particularly Table 3 and Figures 3 and 4, need clearer labels and captions to improve interpretability. For example, in Figure 3, in-figure labels and descriptive legends would make key findings more accessible. Table 3 could be reorganised to highlight the comparison across models more explicitly (possibly making certain results bold for which is the best performing architecture). Table 3, in particular, is very difficult to assess if the proposed model results are competitive.

### Insufficient Background on Key Concepts

- **Explanation of 2D Graph Similarity**: While low 2D graph similarity is cited as a goal, the authors do not provide a rationale for its significance. It is crucial, as the authors do, to explain that minimizing 2D similarity while maximizing 3D shape alignment promotes structural novelty, which is beneficial for discovering new ligands. This discussion should be expanded in the background section to establish a clearer objective for readers unfamiliar with scaffold-hopping.

- **Background on Zero-weighted MLP and Graph ControlNet**: The paper introduces zero-weighted MLPs within Graph ControlNet, which may be unfamiliar to many readers. Including a brief background on this concept would provide context on its role in stabilizing training and gradually integrating shape constraints.

### Weak Conclusions and Reflection on Results

The conclusions section could be enhanced by reflecting more critically on the limitations of Diff-Shape, such as its dependency on a pre-trained model and challenges related to low validity when fuzzy operations are applied. Additionally, providing insights into the real-world applicability of Diff-Shape and potential improvements (e.g., exploring alternative conditioning mechanisms) would strengthen the overall impact.


### Minor

- **GitHub Repository**: The repository breaches anonymity as the authors’ full names and contact details are in the setup.

- **Grammatical, Citation & Spelling Errors**:
  - In-text citation should be before a period.
  - Authors are not consistent in using a space after a comma.
  - There are many occasions of missing articles in sentence structures (e.g., "chosen for doing following experiments in our current study").
  - Incorrect grammar on multiple occasions (e.g., "called as Diff-Shape," "in later case").
  - GEOM-Drugs dataset was not cited.

### Equations

- After equation 5, where the graph embedding, $G^{t+1}$, is defined, $A^{t+1}$ and $C^{t+1}$ are not defined. As a result, $a_{i}^{t-1}$  and $c_{i}^{t-1}$  are not defined either.

- In Equation 8, $\lambda$ is not defined, and it is unclear what kind of weighting is allocated to the "weighted sum of the components" in the loss function. Is it randomized or parameterized?

**Questions:**

1. **Pipeline Sensitivity**: How sensitive is the overall pipeline to the unconditional pre-trained model? The paper provides little information about the rationale behind the use of MIDI and does not assess alternative unconditional models.

---

### Official Review · Reviewer_E27B · 2024-11-02

**Soundness:** 2
**Presentation:** 3
**Contribution:** 2
**Rating:** 3
**Confidence:** 4

**Summary:**

The paper introduces Diff-Shape, a novel diffusion model designed for shape-based virtual screening in ligand-based de novo drug design. Unlike traditional methods that rely on chemical libraries, Diff-Shape directly generates 3D molecular structures guided by a reference shape, thereby reducing the computational costs associated with screening large databases. By employing a zero-weighted graph control module, the model incorporates various point cloud representations of the reference shape to steer the diffusion process, enabling the generation of molecules with high 3D shape similarity and low 2D graph similarity to the target structure.

**Strengths:**

●  	New architecture for effective Shape Guidance: The use of a zero-weighted graph control module effectively guides the diffusion process, allowing for the generation of molecules that maintain high 3D shape similarity.

●  	High Performance: The model demonstrates superior performance compared to existing shape-based generative models, indicating its capability to produce valid and diverse drug-like molecules.

●  	Broad Chemical Space Exploration: By generating novel 3D structures, Diff-Shape enhances the ability to explore larger chemical spaces.

●  	Integration of Existing Models: The ability to incorporate a pre-trained unconditional diffusion model allows for efficient utilization of prior knowledge, facilitating improved outcomes without the need for retraining from scratch.

**Weaknesses:**

●  	Definitions: Please define the terms "validity" and "uniqueness" in the context of your work.

●  	Drug-Likeness of Molecules: The paper does not provide sufficient information on whether the generated molecules adhere to drug-likeness criteria, which is essential for their potential application in drug discovery.

●  	Responsiveness to Shape Variation: There is a lack of analysis regarding how the model's generation is affected by variations in the size of the reference shape. Demonstrating responsiveness in this context would strengthen the model's validity.

●  	Synthesis Feasibility: The paper does not address the synthetic feasibility of the generated molecules. Assessing how easily these compounds can be synthesized is critical for practical applications in drug discovery.

●  	Performance with Rotatable Bonds: There is no insight into how the model performs with varying numbers of rotatable bonds in both the seed and generated compounds. This information is important for understanding the flexibility and potential conformational diversity of the generated molecules.

●  	Limited Insight into Model Behavior: While the work is impressive, the lack of detailed discussion around the aforementioned points raises questions about the model's comprehensive applicability and effectiveness in real-world scenarios.

**Questions:**

Could you provide evidence that you can generate molecules that are diverse in terms of their properties?

---

### Official Review · Reviewer_Fqes · 2024-11-03

**Soundness:** 2
**Presentation:** 2
**Contribution:** 2
**Rating:** 5
**Confidence:** 4

**Summary:**

This study developed a diffusion model that generates 3D molecules conditioned on shape.

It was experimentally demonstrated that the model achieves higher shape similarity and graph similarity compared to existing models that generate 3D molecules conditioned on shape (e.g., Fig 6).

An additional model and retraining technique were proposed to adapt the existing unconditional MIDI model for generating 3D molecules conditioned on shape. As a result, it was shown that the model can generate 3D molecules with higher shape similarity and graph similarity than those produced by the original MIDI model (e.g., Fig 5).

**Strengths:**

- Development of a shape-based 3D molecular generation model through a concise and intuitive additional model design, with open-source code release.

- Demonstrates the add-on capability to existing baselines such as MIDI and presents a unique architecture.
Shows high performance in maintaining 3D shape.

- Provides ablation study results for various input conditions beyond shape.

**Weaknesses:**

Weak Novelty:

- Studies utilizing shape as the input condition to generate 3D molecule already exist, as author mentioned (e.g., SQUID, ShapeMol - see below (1) (2) (3) papers, etc). Therefore, novelty must be highlighted as performance superiority rather than conceptional novelty.

Details :  The authors, inspired by ControlNet, applied its principles to a 3D molecular generation model, proposing a state-of-the-art approach that uses shape as an input to preserve the shape in the generated molecules. While there is a novel aspect in utilizing graph-based ControlNet in the molecular generation domain, the paper lacks clear experimental evidence to justify the necessity of this approach. Furthermore, it is not the first to use shape constraints as input for 3D molecular generation, as several studies have already explored similar approaches. This raises questions about the conceptual novelty, which is not well articulated in the paper. For instance, the paper does not clearly explain how it conceptually differs from existing models that use encoders to interpret shape constraints for 3D molecular generation (see related studies below (1) (2) (3)). The authors acknowledge this and instead emphasize the model’s superior performance compared to other shape-constrained generation models (see lines since 097). However, without clear performance improvements (see weakness 2), the advantages or novelty of this model over other shape-constrained generation models remain unclear.

(1) Equivariant Shape-Conditioned Generation of 3D Molecules for Ligand-Based Drug Design [https://openreview.net/forum?id=4MbGnp4iPQ / https://arxiv.org/abs/2210.04893]

(2) ShapeMol : Shape-conditioned 3D Molecule Generation via Equivariant Diffusion Models [https://arxiv.org/pdf/2308.11890]

(3) Shape-Based Generative Modeling for de Novo Drug Design [https://pubs.acs.org/doi/10.1021/acs.jcim.8b00706]


Unclear Performance Improvement:

- In Figures 5 or 6, while the proposed method significantly increases shape similarity (advantage), it also increases graph similarity (disadvantage). The performance improvement would be validated if shape similarity increased while maintaining graph similarity, but this is not the case, making it ambiguous to claim a performance enhancement.

- Table 2 shows a significant decrease in Validity * Uniqueness compared to MIDI. It is unclear whether the proposed method clearly outperforms MIDI.

- In Table 3, while the result showing low graph similarity for shape similarity above 0.8 is interesting, there is concern about its parametric nature. For instance, if the comparison is based on shape similarity above 0.4 instead of 0.8, the proposed method might exhibit higher graph similarity than other methods.

[Details] In Table 3, the value in the fourth column (P^{Simg<1.0}_{Sim3D>0.8}) seems to best highlight the strengths of this approach. However, this advantage does not appear consistent across different thresholds (e.g., Sim3D > 0.4). Figure 3 shows that graph similarity unfortunately does not consistently increase alongside shape similarity. If the graph similarity had been lower or comparable to existing methods, while shape similarity had been higher, the superiority of this approach would have been clearer. Unfortunately, this is not the case, and thus the paper falls short of providing clear evidence of performance superiority, which may hinder its acceptance at ICLR. Additionally, beyond the limitations in Validity and Uniqueness compared to other shape-constrained generation models in Table 3, the paper fails to demonstrate that the "graph similarity/shape similarity" metric consistently outperforms other methods across thresholds. This makes it difficult to identify the distinct technological advantages of this approach.

- There is a lack of detailed explanation regarding whether the experiments for existing studies (SQUID, ShapeMol) were conducted under fair conditions with the same input conditions and without pre-trained checkpoints. Additionally, the absence of base code makes it difficult to verify this. Performance comparisons under the same conditions as those presented in existing studies should also be provided.

======

Depending on the responses from other reviewers and the authors, if the concerns can be sufficiently addressed, I will consider adjusting the review score accordingly.

**Questions:**

Please refer to my comment for Weaknesses section.

---

### Meta-Review · Area_Chair_gGSH · 2024-12-15

**Metareview:**

This submission presents a diffusion model making use of 3D reference shapes to generate novel molecules. The main claim of the paper is that the additional conditioning results in chemically novel shapes as compared to a reference structure. While the conceptual simplicity of using reference shapes to essentially encourage "plausible" explorations of the molecular space is appreciated, the current version of the manuscript suffers from several weaknesses, to wit:

1. Presentation issues, which preclude a clear understanding of the method.
2. A lack of contextualisation and comparison partners, which would serve to better understand the results.
3. Issues with the experimental setup such as a mixture of different out-of-distribution and in-distribution comparisons.

While some of these issues could have been addressed in a rebuttal, the authors did not supply one, letting the initial criticisms of reviewers stand on their own. In case the authors are interested in improving their paper, from my reading of the discussion and of the paper itself, the most salient points are the *presentation issues*, followed by an improved *experimental setup*.

**Additional Comments On Reviewer Discussion:**

Reviewers raised concerns along different dimensions, including conceptual novelty (`Fqes`, `bACC`), unclear performance improvements (`Fqes`, `E27B`, `bACC`) and a lack of clarity (`zLm1`, `bACC`). Moreover, reviewer `bACC` raised concerns about plagiarism. The SAC and the myself agree with this assessment and have raised the issue to the PCs, as well as to the ethics reviewers. Since no rebuttal was provided, we have to let these allegations stand as they are, but it is unfortunate that the authors did not choose to address at least some of these concerns.

Since none of the points have been addressed in the rebuttal, I agree with the first assessment of the reviewers and will suggest rejecting the paper. I sincerely hope that the authors make use of the ample opportunities for improving their work, which have been pointed out to them in the reviewing process.

---

### Decision · Program_Chairs · 2025-01-22

Reject